# Differential Diagnosis for Highly Pathogenic Avian Influenza Virus Using Nanoparticles Expressing Chemiluminescence

**DOI:** 10.3390/v13071274

**Published:** 2021-06-30

**Authors:** Jihee Kim, Jae-Yeon Park, Jihoon Ryu, Hyun-Jin Shin, Jung-Eun Park

**Affiliations:** 1Laboratory of Veterinary Public Health, College of Veterinary Medicine, Chungnam National University, Daejeon 34134, Korea; wlgml0721@naver.com; 2Laboratory of Veterinary Infectious Diseases, College of Veterinary Medicine, Chungnam National University, Daejeon 34134, Korea; wodus5818@naver.com (J.-Y.P.); jihoon0511@cnu.ac.kr (J.R.); shin0089@cnu.ac.kr (H.-J.S.); 3Research Institute of Veterinary Science, Chungnam National University, Daejeon 34134, Korea

**Keywords:** avian influenza, differential diagnosis, nanoparticle, chemiluminescence

## Abstract

Highly pathogenic avian influenza (HPAI) virus is a causative agent of systemic disease in poultry, characterized by high mortality. Rapid diagnosis is crucial for the control of HPAI. In this study, we aimed to develop a differential diagnostic method that can distinguish HPAI from low pathogenic avian influenza (LPAI) viruses using dual split proteins (DSPs). DSPs are chimeras of an enzymatic split, Renilla luciferase (RL), and a non-enzymatic split green fluorescent protein (GFP). Nanoparticles expressing DSPs, sialic acid, and/or transmembrane serine protease 2 (TMPRSS2) were generated, and RL activity was determined in the presence of HPAI or LPAI pseudotyped viruses. The RL activity of nanoparticles containing both DSPs was approximately 2 × 10^6^ RLU, indicating that DSPs can be successfully incorporated into nanoparticles. The RL activity of nanoparticles containing half of the DSPs was around 5 × 10^1^ RLU. When nanoparticles containing half of the DSPs were incubated with HPAI pseudotyped viruses at low pH, RL activity was increased up to 1 × 10^3^ RLU. However, LPAI pseudotyped viruses produced RL activity only in the presence of proteases (trypsin or TMPRSS2), and the average RL activity was around 7 × 10^2^ RLU. We confirmed that nanoparticle fusion assay also diagnoses authentic viruses with specificity of 100% and sensitivity of 91.67%. The data indicated that the developed method distinguished HPAI and LPAI, and suggested that the diagnosis using DSPs could be used for the development of differential diagnostic kits for HPAI after further optimization.

## 1. Introduction

Avian influenza (AI) is a disease caused by infection with the influenza A virus [1]. AI viruses are typically characterized by low pathogenicity (LP), causing subclinical infections, respiratory disease, or drops in egg production [2]. However, a few AI viruses are highly pathogenic (HP), causing severe systemic disease with multiple organ failure and high mortality [2]. Furthermore, zoonotic infections of HPAI have been reported in humans accompanied by conjunctivitis, respiratory disease, or multi-organ failure and death [3]. For the confirmation of HPAI, laboratory tests, such as virus isolation in embryonated eggs or cultured cells, and detection of viral RNA or antibodies are required [4,5]. Stamping-out programs are used to quickly eliminate HPAI viruses in developed countries, while some developing countries use vaccines and other management strategies to control HPAI viruses.

Influenza A virus is a member of the family *Orthomyxoviridae*, which contains enveloped viruses with segmented RNA genomes of negative polarity. Virus infection is initiated by the binding of hemagglutinin (HA) to cell surface receptors, sialic acids [6]. Following cell entry through endocytosis, the HA proteins change conformation under low pH conditions in the endosome, which causes fusion of viral-endosomal membranes [6]. For membrane fusion to occur, the precursor protein HA0 must become proteolytically cleaved into HA1 and HA2 subunits in the trans-Golgi network or on the plasma membrane [7]. In contrast with the HA proteins of LPAI, the HA proteins of HPAI viruses contain multiple basic amino acids at the cleavage site [8]. This characteristic sequence is cleaved by ubiquitous cellular proteases, such as furin and proprotein convertase [9], suggesting systemic infection in poultry. Thus, the amino acid sequence of the HA cleavage site is a key determinant of organ tropism and pathogenicity of influenza A viruses [10,11].

Rapid detection and accurate identification of HPAI are crucial for the control of HPAI infection in poultry [12]. Various diagnostic methods have been developed [13]. Although fast, rapid diagnostic kits cannot distinguish between low and high pathogenic viruses [14,15,16]. Therefore, additional laboratory tests, such as reverse transcriptase polymerase chain reaction (RT-PCR), sequencing, and virus isolation in embryonated eggs or cultured cells are necessary for the confirmation of HPAI [4,5]. However, these methods are time-consuming, expensive, and require sophisticated equipment and skills. To date, there have been no rapid and confirmative methods for HPAI diagnosis.

Dual split protein (DSP) is a chimeric reporter protein composed of split renilla luciferase (RL) and split green fluorescence protein (GFP) [17]. DSP has almost no activity but expresses both RL and GFP after re-association [17]. This is capable for various biological events such as membrane fusion and protein–protein interaction. This quick assay provides quantitative as well as spatial information about membrane fusion mediated by viral and cellular fusion proteins, especially in cell–cell fusion assay [18,19,20,21]. In the present study, we developed a nanoparticle fusion assay using HIV pseudotyped nanoparticles containing DSPs. The nanoparticle fusion assay was able to detect AIV with high specificity and sensitivity. Our data prove that the assay is a useful field diagnostic tool for rapid quarantine control of AIV.

## 2. Materials and Methods

### 2.1. Cells

HEK293T cells were maintained in Dulbecco’s modified Eagle’s medium (DMEM) supplemented with 10% (*v*/*v*) fetal bovine serum (FBS), 10 mM 4-(2-hydroxyethyl)-1-piperazineethanesulfonic acid (HEPES), 100 mM sodium pyruvate, 0.1 mM nonessential amino acids, 100 U/mL penicillin G, and 100 μg/mL streptomycin. Huh7 cells were maintained in DMEM supplemented with 10% (*v*/*v*) FBS, 100 U/mL penicillin G, and 100 μg/mL streptomycin. Cell culture materials and reagents were obtained from SPL Life Sciences Co., Ltd. (Pocheon, Korea) and Hyclone (HyClone Laboratories Inc., South Logan, UT, USA), unless otherwise noted.

### 2.2. Plasmids

Recombinant pCAGGS-S15-DSP_1–7_ and pCAGGS-S15-DSP_8–11_ were constructed by fusing the DSP_1–7_ or DSP_8–11_ coding sequences (pDSP1–7 and pDSP8–11 [18] provided by Zene Matsuda (University of Tokyo)) the N-terminal 15 amino acid sequence of c-Src (S15), and provided by Tom Gallagher, Loyola University Chicago. pCAGGS-TMPRSS2-Flag was previously described [22]. pcDNA3.1-HA5-QH, pcDNA3.1-HA5-QH-trypsin site, and pcDNA3.1-PR8 NA1 were provided by Lijun Rong, University of Illinois-Chicago and was previously described [23]. The pNL4.3-HIVluc (luc stands for firefly luciferase) plasmid was provided by the NIH AIDS Research and Reference library.

### 2.3. Generation of HIV Nanoparticles

For production of nanoparticles expressing DSPs, HEK293T cells were transfected with pNL4.3-HIVluc in conjunction with plasmids encoding S15-DSP_1–7_ or S15-DSP_8–11_. Where indicated, TMPRSS2-Flag was co-transfected. For the production of pseudotyped influenza viruses, HEK293T cells were transfected with pNL4.3-HIVluc and pcDNA3.1-PR8 NA in conjunction with plasmids encoding pcDNA3.1-HA5-QH or pcDNA3.1-HA5-QH-trypsin site. Transfection was performed by incubating plasmid DNAs with polyethylenimine (PEI) at 1:3 DNA:PEI ratios in Opti-MEM (Life Technologies, Carlsbad, CA, USA) for 15 min at 25 °C. Cell-free supernatants containing the nanoparticles were collected at 48 h post-transfection, filtered through 0.45-µm syringe filters (Pall Life Sciences, Port Washington, NY, USA), and stored at −80 °C until use.

### 2.4. Pseudotyped AIV Preparation and Transduction

For the production of pseudotyped influenza viruses, HEK293T cells were transfected with pNL4.3-HIVluc and pcDNA3.1-PR8 NA in conjunction with plasmids encoding pcDNA3.1-HA5-QH or pcDNA3.1-HA5-QH-trypsin site. Transfection was performed as described above. Cell-free supernatants containing the pseudotyped viruses were collected at 48 h post-transfection, filtered through 0.45-µm syringe filters (Pall Life Sciences, New York, NY, USA), and stored at −80 °C until use. 

Huh7 cells were transduced with pseudotyped influenza viruses for 1 h at 37 °C, washed, and further incubated for an additional 48 h. At 48 h post-transduction, cells were dissolved in cell culture lysis buffer (Promega, Madison, WI, USA) and relative luciferase units (RLU) were measured by the addition of Firefly luciferase substrate (Promega) using a GloMax^®^ Navigator Microplate Luminometer (Promega).

### 2.5. Determination of DSP Expression

Transfected cells were dissolved in cell culture lysis buffer, and luciferase levels were measured by the addition of RL substrate (1 mM D-luciferin, 3 mM ATP, 15 mM MgSO4·H2O, and 30 mM HEPES [pH 7.8]) using a GloMax^®^ Navigator Microplate Luminometer (Promega). For GFP expression, cells were fixed with 4% paraformaldehyde for 20 min at 25 °C. The cells were imaged with a 4 Cubes Fluorescence Microscope (Optinity KI-2000F, Korea Lab Tech, Seongnam, Korea) equipped with a digital camera (Optinity KCS3–63S, Korea Lab Tech, Seongnam, Korea).

For RL detection of nanoparticles, nanoparticles were incubated with ViviRen™ LiveCell Substrate (Promega) in the same volume for 10 min at 25 °C. RL activity was measured using GloMax^®^ Navigator Microplate Luminometer.

### 2.6. Western Blots

Nanoparticles were incubated with neuraminidase A (Neu A, New England biolabs, Inc., Beverly, MA, USA), neuraminidase S (Neu S, New England biolabs, Inc.), or PNGase F (New England biolabs, Inc.) at 37 °C for 18 h. Proteins were transferred to a polyvinylidene fluoride membrane (Thermo Fisher Scientific, Waltham, MA, USA). After blocking with 5% (*w*/*v*) bovine serum albumin in Tris-buffered saline with 0.05% TWEEN 20 (TBST) for 1 h at 25 °C, membranes were probed with monoclonal mouse anti-flag (1:1000, Sigma-Aldrich, St. Louis, MO, USA) antibodies, biotinylated Sambucus nigra agglutinin (SNA, Vector Laboratories, Burlingame, CA, USA), or biotinylated Maackia amurensis lectin-I (MAL-I, Vector Laboratories, Burlingame, CA, USA) as a primary staining, and then with horseradish peroxidase-conjugated goat anti-mouse IgG (Bioss Antibodies, Woburn, MA, USA) and streptavidin at 1:5000 dilution in TBST. Membranes were developed using ECL substrate (Thermo Fisher Scientific, Waltham, MA, USA) and signals were detected with Fusion Solo X (Vilber, Paris, France).

### 2.7. Nanoparticle Fusion Assay

Nanoparticles were incubated with pseudotyped influenza viruses, authentic viruses, or fecal samples in a ratio of 1:3 for 1 h at 4 °C. Authentic influenza viruses (H1N1 and H9N2) and other avian viruses (Newcastle disease virus [NDV], infectious bursal disease virus [IBDV], and infectious bronchitis virus [IBV]) were obtained from Hyun-Jin Shin at Chungnam National University (South Korea). Nanoparticles were then incubated with pH buffered solution (pH4.6, made by 0.1 M citric acid with 0.2 M Na_2_HPO_4_) in a ratio of 1:5 for 10 min at 25 °C. Nanoparticle fusion was analyzed by the RL assay as described above.

### 2.8. Fecal Samples

A total of 103 fecal samples from wild-bird habitats were collected in three areas in South Korea. Fecal samples were examined by RT-PCR using following primers; forward primer 5′-ATGAGTCTTCTAACCGAGGTCGAAAC-3′ and reverse primer 5′-CTTGAATCGTTGCATTTGCACTCC-3′.

### 2.9. Statistical Analysis

All experiments were independently repeated at least three times. Data are presented as mean ± SD. Statistical significance was calculated using the Holm–Sidak multiple Student’s t test. A *p* value of <0.05 was considered statistically significant.

## 3. Results

### 3.1. Generation of Nanoparticles Expressing DSPs

HIV pseudotyped nanoparticles containing DPSs were generated. To increase the integration of DSPs into nanoparticles, S15-DSPs were engineered. It was previously shown that fluorescent protein fusion targeted to the plasma membrane by the addition of the N-terminal 15 amino acid sequence of c-Src is efficiently packaged into HIV virions [24]. HEK293T cells were transfected with S15-DSPs, and RL activity was determined (Figure 1A). Similar to previous reports, basal level of RL activity was slightly higher in DSP_8-11_ transfected cells than in DSP_1–7_ transfected cells. When cells were co-transfected with both DSPs, RL activity was significantly increased. To examine the subcellular localization of S15-DSPs, GFP expression was monitored. No GFP signal was detected in cells transfected with each DSP. GFP was detected in cells co-transfected with both DPSs, particularly in the plasma membrane (Figure 1B).

To generate nanoparticles expressing DSPs, HEK293T cells were transfected with plasmids encoding S15-DSPs and HIV core proteins. Cell-free supernatants were harvested, and nanoparticles were purified by ultracentrifugation. Basal levels of RL activity were detected in the nanoparticles obtained from cells transfected with each DSP. RL activity was significantly higher in the nanoparticles obtained from cells transfected with both DPSs (Figure 1C). Our results indicated that DSPs could be integrated and self-assembled in nanoparticles.

### 3.2. HPAI Induced Nanoparticle Fusion at Low pH

HPAI HA binds to sialic acids and induces membrane fusion at low pH. We examined whether HPAI HA induces nanoparticle fusion upon low pH stimulation. Before testing nanoparticle fusion, we first confirmed sialic acid expression in the nanoparticles (Figure 2A). The presence of α2,3 and α2,6-sialylated glycans was supported by staining with commonly used sialic acid-recognizing lectins, MAL-I and SNA, respectively [25,26]. MAL-I bindings were diminished by PNGase F, Neu S, and Neu A treatment. SNA bindings were diminished by PNGase F and Neu A treatment. Pseudotyped viruses expressing HPAI HA, LPAI HA, or no glycoproteins (BALD) were generated, and the transduction ability of pseudotyped viruses was determined in Huh7 cells. The RLUs per ml of HPAI and LPAI pseudotyped viruses were approximately 10^6^ RLU/mL, respectively (data not shown). Nanoparticles were mixed with pseudotyped viruses, and the pH was adjusted to five. Upon low pH stimulation, RL activity was detected in nanoparticles mixed with HPAI pseudotyped viruses, but not in nanoparticles mixed with LPAI or BALD pseudotyped viruses (Figure 2B). When LPAI was treated with trypsin, LPAI was able to induce nanoparticle fusion. Our results indicated that HPAI HA, but not LPAI HA, could induce the fusion of nanoparticles containing sialic acids.

### 3.3. Addition of Proteases Induced LPAI-Mediated Nanoparticle Fusion

Unlike HPAI, HA cleavage by host proteases is essential for LPAI infection. We hypothesized that nanoparticles expressing DSPs would detect LPAI if proteases were present. To test our hypothesis, we generated nanoparticles expressing type II transmembrane serine protease (TMPRSS2). TMPRSS2 is capable of activating HA proteins possessing a single arginine residue at the cleavage site. Using Western blot, we confirmed sialic acid and TMPRSS2 expression in the nanoparticles (Figure 3A). Nanoparticles were incubated with HPAI or LPAI pseudotyped viruses. HPAI pseudotyped viruses showed RL activity regardless of the presence of TMPRSS2, whereas LPAI pseudotyped viruses showed RL activity only in the presence of TMPRSS2 (Figure 3B). To investigate the sensitivity of the assay, serially diluted pseudotyped viruses were diagnosed using nanoparticle fusion assay. HPAI pseudotyped viruses in the 10^3^ to 10^6^ RLU/mL range were positive in nanoparticle fusion assay (Figure 3C). LPAI pseudotyped viruses showed similar sensitivity in the presence of TMPRSS2, whereas no RL activity was observed in the absence of TMPRSS2 (Figure 3C). Our results indicated that LPAI HA could induce nanoparticle fusion only in the presence of proteases.

### 3.4. Nanoparticles Fusion Assay Detected Authentic AI

Finally, we tested whether authentic AI also induced nanoparticle fusion. As expected, both tested LPAI (H9N2 and H1N1) showed RL activity only in the presence of TMPRSS2 (Figure 4A). To investigate the sensitivity of the assay, serially diluted H9N2s were diagnosed using nanoparticle fusion assay. All H9N2s in the 5 to 5 × 10^5^ EID_50_/_mL_ range were positive in nanoparticle fusion assay (Figure 4B). The RL assay was negative for the samples including NDV, IBV, and IBDV (Figure 4). Our results indicate that the nanoparticle fusion assay have high specificity and sensitivity.

We further evaluated the veterinary performance of the assay and compared it with RT-PCR in 103 avian fecal samples obtained from wild poultry (Table 1). RT-PCR analysis detected 12 of the 103 samples as positive (11.65%), while the nanoparticle fusion assay identified and differentiated 11 positive samples with a sensitivity of 91.67 and 100% specificity. The κ value for RT-PCR and the nanoparticle fusion assay was 0.951. All tested fecal samples were negative in the nanoparticle fusion assay with nanoparticles without TMPRSS2 (data not shown)

All the detected positive fecal samples were verified as LPAI by conventional PCR and sequence analysis. Thus, we further validated whether nanoparticle fusion assay diagnose HPAI in fecal samples. To do that, HPAI pseudotyped viruses were mixed with fecal samples and nanoparticle fusion assay was performed. As expected, fecal samples only showed positive reactions in nanoparticle fusion assay with nanoparticles containing TMPRSS2 (Figure 5). When HPAI pseudotyped viruses were added, RL activity was detected in both conditions with or without TMPRSS2 (Figure 5). The data collectively indicated that nanoparticle fusion assay allowed the specific detection of HPAI in fecal samples.

## 4. Discussion

In summary, here we described a new diagnostic method that can detect AI infection. This method distinguished between low and high pathogenicity by determining the requirements for HA-induced membrane fusion. Reporter proteins used in the assay were rapidly assembled and showed detectable expression. Therefore, the developed method allowed rapid and confirmative detection of HPAI.

Rapid detection of the HPAI viruses in the field is crucial to the effective control of AI outbreaks [27,28]. Current rapid diagnostic tests have been developed for the rapid diagnosis of influenza virus infection in point-of-care settings. These tests use monoclonal antibodies that target the viral proteins and employ either enzyme immunoassay or immune-chromatographic (lateral flow) techniques [14,15,16]. The results are visually observed based on color changes or other optical signals in less than 30 min. Owing to the simplicity of their use and the speed of results, rapid diagnostic tests are commonly used for the diagnosis of influenza infections. Most of these tests can either detect or distinguish influenza viruses, whereas, none of them can distinguish between different influenza A subtypes and its pathogenicity [29]. Therefore, HPAI is confirmed by laboratory diagnostic methods such as PCR. However, these methods require complex, expensive devices and are difficult to perform in the field.

Here, we developed nanoparticle fusion assay for rapid, sensitive, and specific detection of the HPAI. Our method is a novel assay requiring relatively short time (~60 min) to complete, compared with approximately 3 h for PCR. In PCR, additional steps such as viral isolation was required. Samples can be directly used for our assay without any further preparation. In addition, the assay steps are simple and easy to do. Therefore, this test method has the advantage that it is as simple and fast as a simple diagnostic kit, and it allows accurate diagnosis such as laboratory diagnostic methods such as PCR.

We and other researchers propose that enzyme specificity is a characteristics that enables specific diagnostics of HPAI. Recently, Kim et al., also reported the rapid detection method for HPAI using host cell-mimicking polymersomes [30]. They generated synthetic nanoparticles containing fluorescence resonance energy transfer pair fluorophores (DiO and DiI). This assay is similar to our assay in that is exploits the enzyme specificity that cleaves that HA proteins. Docking occurred between polymersomes and the fusion peptide in influenza HA activated in the presence of proteases (furin and/or trypsin). This docking allowed for rapid distinction between HPAI and LPAI. Proteases activity can be blocked by various components of the sample, such as serum. Conversely, excess proteases can degrade viral proteins. Therefore, it is thought to be important to precisely control proteases activity.

Sensitivity is important for the performance of point-of-care diagnostics due to the wide ranges of viral titer in field samples. The detection limits of our assay were around 5 EID_50_/_mL_ (Figure 4B, for authentic viruses) and 10^3^ RLU/mL (Figure 3C, for pseudotyped viruses). However, the RL signal was relatively low compared to that of nanoparticles containing both DSPs. We speculate that the low signal is caused by randomly arranged nanoparticles. The sensitivity of the assay would be increased by cross-linking nanoparticles with DSP_1–7_ to nanoparticles with DSP_8–11_.

Sialic acids are a family of nine-carbon acidic monosaccharides that occur naturally at the end of sugar chains attached to the surfaces of cells and soluble proteins. The kind of linkage is of great importance since different HA subtypes have different preferences in binding to one of these sialic acids linkages. Whereas avian influenza HAs preferentially bind to α-2,3 sialic acids, human adapted HA subtypes have been shown to favor α-2,6 sialic acids [31,32,33]. Our Western blot results indicated that nanoparticles expressed both α-2,3 and α-2,6 sialic acids (Figure 2A and Figure 3A), suggesting that the developed method can be used for both avian and human influenza viruses. Indeed, nanoparticle fusion was observed in the presence of human influenza virus H1N1 as well as avian influenza virus H9N2 (Figure 4A). Therefore, the assay could diagnose the other influenza viruses (such as human influenza virus and swine influenza virus), and provide a powerful and valuable tool for the control of influenza virus.

Until now, diagnosis of influenza is based on the detection of the gene of the virus or the detection of antibodies to the viral protein. Our diagnosis is based on the mechanism by which viruses infect cells. Thus, the current study provides the first report with novel mechanism.

In conclusion, we developed a new diagnostic method that can rapidly and differentially detect HPAI. The primary advantages of this innovative method are its fast, easy, and confirmative approach. The development of a rapid diagnostic kit using this method will allow on-site HPAI diagnosis. However, it seems necessary to find the optimal conditions for improving the signal of nanoparticle fusion. Moreover, there seems to be a need to diagnose other viruses known to infect poultry that can confirm the specificity of this diagnostic method.

## Figures and Tables

**Figure 1 viruses-13-01274-f001:**
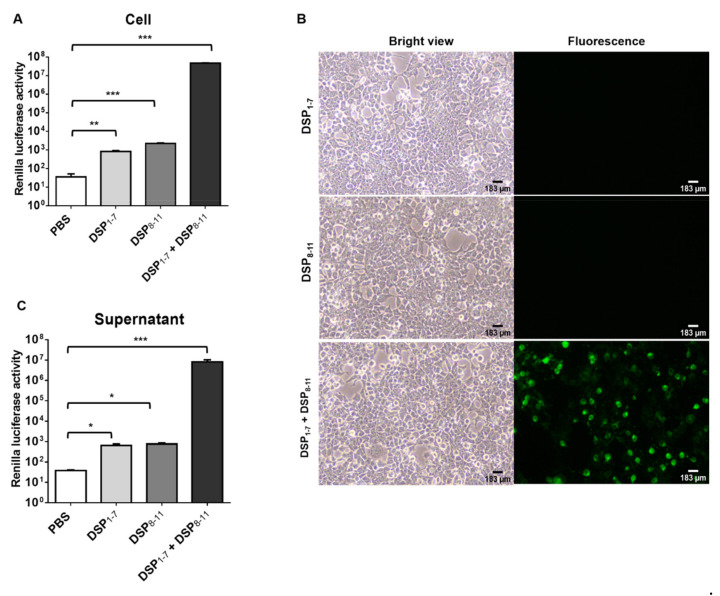
Generation and validation of nanoparticles expressing DSPs. (**A**,**B**) HEK293T cells were transfected with indicated plasmids. DSP expression was analyzed by measuring RL activity (**A**) and visualizing GFP expression (**B**). (**C**) Nanoparticles expressing DSPs were generated in HEK293T cells and analyzed by measuring RL activity. Error bars present SD from the mean (*n* = 3). Statistical significance was assessed by Student’s t test. *, *p* < 0.05; **, *p* < 0.01; ***, *p* < 0.001.

**Figure 2 viruses-13-01274-f002:**
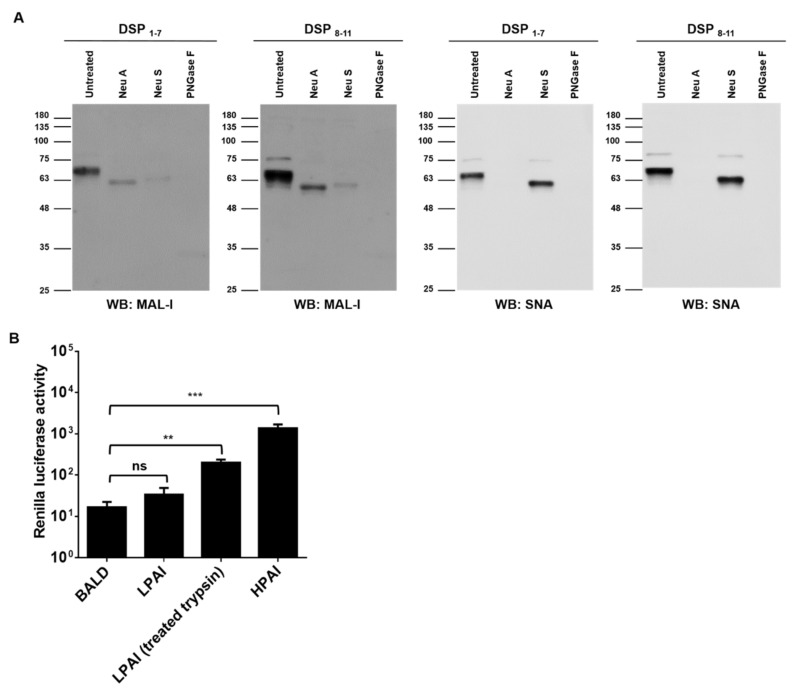
HPAI induced nanoparticle fusion at low pH. (**A**) Nanoparticles expressing DSPs and sialic acids were grown in HEK293T cells and incubated with neuraminidase A (Neu A), neutraminidase S (Neu S), or PNGase F for 18 h at 37 °C. Sialic acids expressions were detected using biotinylated SNA or biotinylated MAL-1. (**B**) Nanoparticles expressing DSPs and sialic acids were incubated with HPAI, LPAI, or BALD pseudotyped viruses for 1 h at 4 °C. Then, mixtures were adjusted at pH 5.0 and incubated with RL substrate for 10 min. Nanoparticle fusion was quantified by measuring RL activity. Error bars present SD from the mean (*n* = 3). Statistical significance was assessed by Student’s t test. **, *p* < 0.01; ***, *p* < 0.001; ns, not significant.

**Figure 3 viruses-13-01274-f003:**
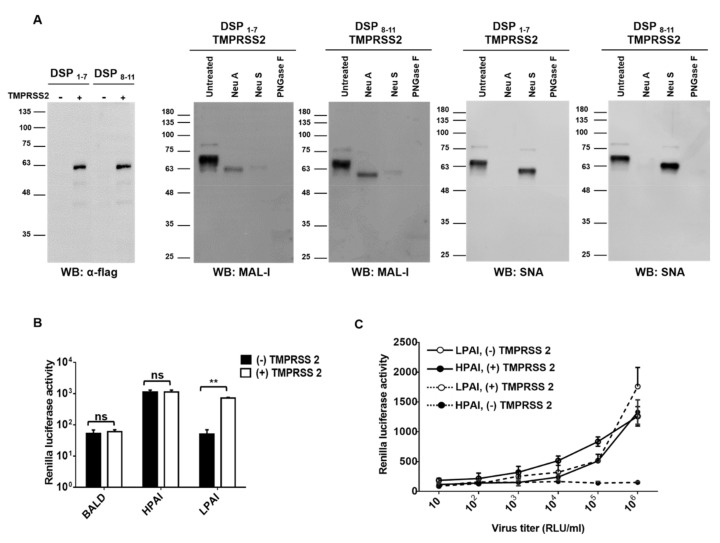
Additional proteases induced LPAI-mediated nanoparticle fusion. (**A**) Nanoparticles expressing DSPs, sialic acids, and TMPRSS2 were grown in HEK293T cells and incubated with neuraminidase A (Neu A), neutraminidase S (Neu S), or PNGase F for 18 h at 37 °C. Sialic acids expressions were detected using biotinylated SNA or biotinylated MAL-1. TMPRSS2 expression was detected using anti-flag antibody. (**B**) Nanoparticles expressing DSPs, sialic acids, and/or TMPRSS2 were incubated with HPAI, LPAI, or BALD pseudotyped viruses for 1 h at 4 °C. (**C**) HPAI and LPAI pseudotyped viruses were 10-fold serially diluted and then incubated with nanoparticles for 1 h at 4 °C. In all experiments, mixtures were adjusted at pH 5.0 and incubated with RL substrate for 10 min. Nanoparticle fusion was quantified by measuring RL activity. Error bars present SD from the mean (*n* = 3). Statistical significance was assessed by Student’s t test. **, *p* < 0.01; ns, not significant.

**Figure 4 viruses-13-01274-f004:**
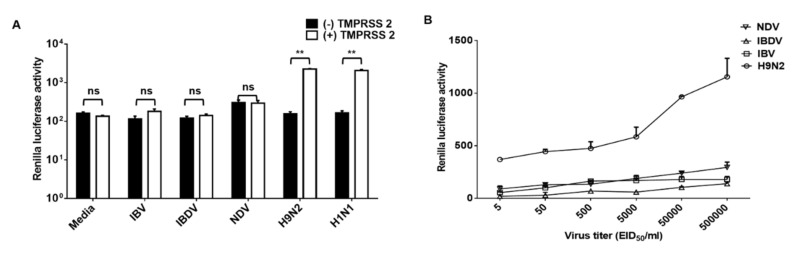
Nanoparticle fusion induced by authentic influenza virus. (**A**) Nanoparticles were incubated with avian influenza virus (H9N2 and H1N1) or avian respiratory viruses (NDV, IBV, and IBDV) for 1 h at 4 °C. (**B**) H9N2, NDV, IBV, and IBDV were 10-fold serially diluted and then incubated with nanoparticles for 1 h at 4 °C. In all experiments, mixtures were adjusted at pH 5.0 and incubated with RL substrate for 10 min. Nanoparticle fusion was quantified by measuring RL activity. Error bars present SD from the mean (*n* = 3). Statistical significance was assessed by Student’s t test. **, *p* < 0.01; ns, not significant.

**Figure 5 viruses-13-01274-f005:**
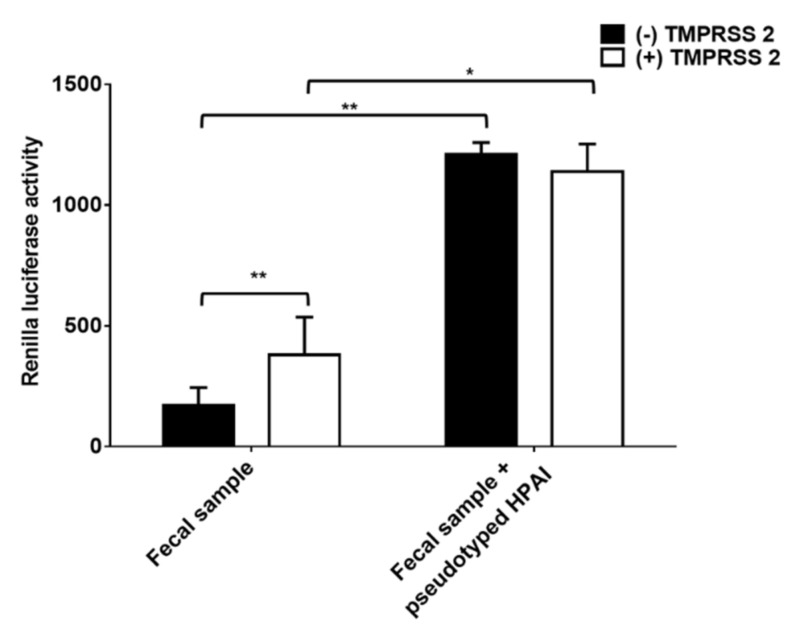
Nanoparticle fusion induced by fecal samples mixed with pseudotyped HPAI. Equal amounts of fecal samples and pseudotyped HPAI were mixed. Nanoparticles were incubated with mixed samples for 1 h at 4 °C. Mixtures were adjusted at pH 5.0 and incubated with RL substrate for 10 min. Nanoparticle fusion was quantified by measuring RL activity. Error bars present SD from the mean (*n* = 8). Statistical significance was assessed by Student’s t test. *, *p* < 0.05; **, *p* < 0.01.

**Table 1 viruses-13-01274-t001:** Detection of AIV in avian fecal samples.

	RT-PCR		Total	Sensitivity (%)	Specificity (%)	Accuracy (%)
	Positive	Negative				
**Nanoparticle fusion assay**						
**Positive**	11	0	10			
**Negative**	1	91	92			
**Total**	12	91	103	91.67%	100%	99.03%

## Data Availability

Not applicable.

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
