# Peer review of "Differential Diagnosis for Highly Pathogenic Avian Influenza Virus Using Nanoparticles Expressing Chemiluminescence"

_viruses, 2021, doi:10.3390/v13071274_

Round 1
Reviewer 1 Report
This paper developed the HPAI virus detection method using nanoparticle with chemilueminescence. They provided the useful data for AIV diagnosis construction. The manuscript may be eventually publishable, but still it requires revision as indicated.
Also, there are some questions raised by reviewer
- For expressing DPSs, what kind of nanoparticle used? Authors should clarify the nanoparticle.
- There is no information of nanoparticle, please show the nanoparticle information and conjugation data such as SEM or others
- All figures resolution qualities are poor. Please have it with high resolution for readers.
Reviewer 2 Report
Avian influenza is an important economical and public concern. The authors developed a differential diagnostic method that can distinguish HPAI from LPAI viruses using dual split proteins. It will be useful tool for rapid diagnosis in the outbreak situation in the poultry as well as in human cases. The manuscript seems to be worthy to be published, however, several points should be addressed before that.
Major points
This diagnosis method is only mixing with nanoparticles and influenza virus in low-pH buffer and determine luciferase activity. Therefore, validation of quality and quantity of nanoparticles is an important point for results. The authors showed high luciferase activity as 108 in figure 1C, however in the actual analysis using pseudotyped virus and influenza virus, positive luciferase activity is only 1 log higher than negative samples. The authors need to validation or discussion about this low reactivity.
The authors used DSPs that key system for this novel diagnostic method, and it is already published method and used for several assays. However, the system did not mention in the introduction. The DSPs system should be explain in the introduction.
Minor points
P2 L75-76, please add references.
P2 L82-85, The plasmid names “S14-DSP1-7 , S15-DSP8-11, and pNL4.3-Luc” are same materials as “DSP1-7 and DSP8-11, and pNL4.3-HIVluc” in 2.2 Plasmids section? If they are same plasmids, you need to use common name for all section.
P2 L87-91, Since the previous sentence explain for production of pseudtyped influenza virus, it is unclear that this transfection and purification method is for only nanoparticle or both for nanoparticle and pseudotyped influenza virus. It would suggest separating explanation about nanoparticle and pseudotypes influenza virus in the section.
P3 L123-124, Virus names should be introduced in the formal name before using abbreviation.
P5 L171-172, The authors said data not shown, and did not explain about transduction ability. The results need to explain in the method or results section if the authors decided the limit for use.
P7 Figure 4B, a little bigger symbol is better to identify each sample.
P7 L224-225, the samples are avian clinical samples obtained from wild poultry, but the authors explain “fecal samples from wild-bird habitats” in the methods section. Are these samples wild-bird or clinical sample in the poultry?
P7 Table 1, Nanoparticle fusion assay should put above “positive”.
P7 results of table 1, If you validate this system using clinical samples, both TMPRSS2 (+) and TMPRSS2 (-) systems should be used for detection to determine of specificity even all samples are LPAI. And it is more effective if a HPAI sample is added for validation. For example, fecal samples mixing with pseudotypes HPAI may mimic HPAI sample if the authors can not access to HPAI samples.
Round 2
Reviewer 1 Report
Still question 2 unsolved. Please carry out nanopartilr analysis using TEM or AFM, others
Author Response
Thank you for your comments. Unfortunately, we do not have the equipment for EM analysis, and we are in a situation whether we cannot perform EM analysis within the review period (5 days). However, as HIV virus-like (or pseudotyped) particles are a system used for many years in worldwide, it is thought that verification through EM is not need. Since EM analysis is not possible, the generation of nanoparticles was verified by transduction assay. Our nanoparticles express sialic acids and DSPs, and also contain genes for firefly luciferase (Fluc) expression. As shown in attached figure, nanoparticles expressing sialic acids (DSP nanoparticle) successfully transduced HA-expressing cells. I hope the answer is acceptable.
